# Risk-Sensitive Rear-Wheel Steering Control Method Based on the Risk Potential Field

**Toshinori Kojima and Pongsathorn Raksincharoensak ***

Department of Mechanical Systems Engineering, Tokyo University of Agriculture and Technology,
2-24-16 Naka-cho Koganei, Tokyo 184-0012, Japan; s200112w@st.go.tuat.ac.jp
* Correspondence: pong@cc.tuat.ac.jp; Tel.: +81-42-388-7397

**Abstract:** Various driving assistance systems have been developed to reduce the number of automobile accidents. However, the control laws of these assistance systems differ based on each situation, and the discontinuous control command value may be input instantaneously. Therefore, a seamless and unified control law for driving assistance systems that can be used in multiple situations is necessary to realize more versatile autonomous driving. Although studies have been conducted on four-wheel steering that steers the rear wheels, these studies considered the role of the rear wheels only to improve vehicle dynamics and not to contribute to autonomous driving. Therefore, in this study, we define the risk potential field as a uniform control law and propose a rear-wheel steering control system that actively steers the rear wheels to contribute to autonomous driving, depending on the level of the perceived risk in the driving situation. The effectiveness of the proposed method is verified by a double lane change test, which is performed assuming emergency avoidance in simulations, and subject experiments using a driving simulator. The results indicate that actively steering the rear wheels ensures a safer and smoother drive while simultaneously improving the emergency avoidance performance.

**Keywords:** vehicle dynamics; autonomous driving; four-wheel steering (4WS); risk potential field





## 1. Introduction

Autonomous driving systems have been practically used for normal driving as well as emergency avoidance driving to realize a safe and comfortable traffic system. Several autonomous driving systems, such as the lane keep assist system and automatic emergency steering, already exist in practical use. However, the control laws of these driving assistance systems differ in each situation. To achieve more versatile autonomous driving, it is necessary to develop a driving assistance system that can be designed with a uniform control law for both normal driving and emergency avoidance driving.

Therefore, the risk potential field is used as a uniform control law in this study. There are many classic risks in vehicle dynamics control, such as sideslip, rollover, and collision. In a previous study, a rollover prevention system using a steer-by-wire system was proposed [1]. In this study, a risk potential field is used to reduce the collision risk among possible risks in vehicle control. Several studies have investigated this control method and applied it to various situations, such as highway merging scenes, emergency avoidance scenes, and shared control [2–6]. As a previous study in our research group, Sato et al. applied a dynamic risk potential to a highway merging scene, wherein the risk potential changed based on the risk of road deviations. The system controls the steering angle, acceleration, and deceleration to drive safely without colliding with other vehicles or road boundaries. The simulation results showed that the dynamic risk potential could reproduce merging behavior identical to that of an expert driver [3]. As an another example of previous work, Inoue et al. applied the risk potential field to a shared control: a driver assistance system with both the human and machine controlling the vehicle. Herein, the driver and machine cooperate to ensure safe driving. They developed motion

planning for obstacle avoidance based on the risk potential field and proposed a control method that combined steering torque control with the direct yaw moment control. The results confirmed an improvement in the steering cooperativity between the driver and machine [4]. This series of papers describes how active chassis control technologies can potentially improve the safety of a vehicle in lane keeping and obstacle avoidance driving scenarios while keeping good cooperative driving between the driver and the machine. However, the control inputs are torque to the steering wheel or the braking and driving force to the tires in these control laws, and they do not control the rear steering angle.

Nevertheless, several studies have explored four-wheel steering (4WS) that steers the rear wheels [7]. A study compared the performance of zero-sideslip angle 4WS and no phase lag in lateral acceleration 4WS [8]. Recently, 4WS equipped with actuators has been practically implemented. Hirose et al. developed a 4WS system that improves turning performance at low velocity and vehicle stability at high velocity without affecting the comfort of the driver [9]. However, in these studies, the rear wheels only improved the vehicle dynamics and did not contribute to autonomous driving. Alleyne proposed the vehicle control in obstacle avoidance situation and compared the performance of various active chassis control systems [10]. Eckert et al. studied obstacle avoidance performance using evasive steering maneuvers, including rear steering control [11]. Deng et al. proposed the control of a four-wheel independent steer-by-wire system for robust steering command tracking [12,13]. Recently, Rau et al. presented the control of rear axle steering in a long-wheelbase production vehicle which can be used for improving handling dynamics and automated parking [14]. In this research, full automation using four-wheel steering control is discussed, while the cooperative driving and characteristics between the human and the system are still research issues.

To address this limitation, in this study, we define the risk potential field for emergency avoidance driving based on the road environment information obtained from onboard environment perception sensors as well as the infrastructure or connected vehicle technology, if available. Additionally, a unified control law of a rear-wheel steering control system is proposed that contributes to autonomous driving for active safety functions. The effectiveness of the proposed method is verified via simulations and an experimental study using a driving simulator of a double lane change considering emergency avoidance driving. The contribution of this work in the field of advanced driver assistance systems and automated driving is to show the new function of the active chassis control, (i.e., active rear steering control) to support the driver in risky driving situations. The level of control intervention using the active chassis control can be adapted, depending on the risk level of the driving situation in which good cooperative driving characteristics between the human and the system (machine) can be achieved.

## 2. Design of Rear-Wheel Steering Control Based on the Risk Potential Field

Figure 1 illustrates the block diagram of the 4WS control system. The system generates a risk potential field based on the road environment information and the vehicle's state, and the reference yaw rate is calculated considering the risk potential. The desired rear-wheel steering angle is calculated using a controller comprising feedforward and state feedback controllers. The system was designed to vary the amount of assistance by changing the weighting coefficient $w$ according to the risk level. However, as an emergency avoidance situation was assumed in this study, we set $w = 1$, and the amount of assistance was the maximum.

In the figure, $X$ and $Y$ indicate the vehicle position, $r$ denotes the yaw rate, $V$ represents the vehicle velocity, $\psi$ indicates the yaw angle, $a_x$ denotes the longitudinal acceleration, $\beta$ represents the vehicle sideslip angle, $U_{risk}$ indicates the value of the risk potential, $r^*$ denotes the reference yaw rate, $\delta_{r\_ff}^*$ and $\delta_{r\_fb}^*$ denote the feedforward and feedback terms of the desired rear steering angle, respectively, $\delta_r^*$ represents the desired rear steering angle, $\delta_{sw}$ indicates the steering wheel angle, $\delta_f$ denotes the front steering angle, and $n$ represents the steering gear ratio.

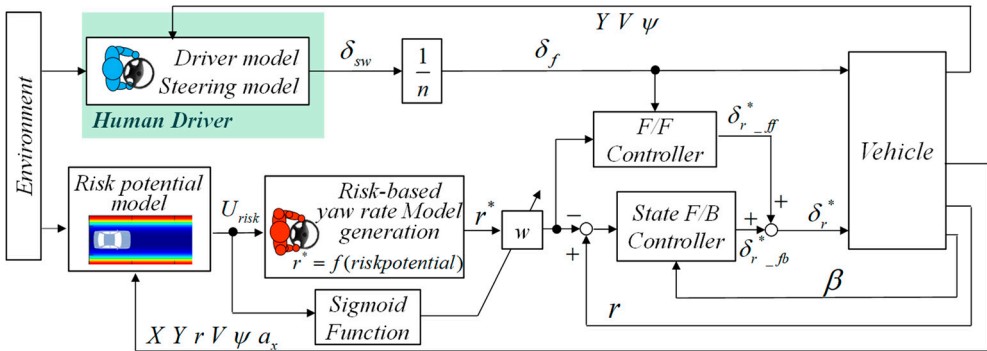

**Figure 1.** Block diagram of the four-wheel steering (4WS) control system.

### 2.1. Design of the Risk Potential Field

As this study aims to design the collision avoidance system, "risk" in this paper refers to the collision risk for the obstacle and the lane boundary. When the relative distance between the vehicle and those objects are closer, the risk becomes higher. This is based on the concept of collision avoidance explained by Reichardt and Schick [15] and the intuitive feeling of the driver's physiological status studied by Kageyama, which explained that the heart rates of drivers get higher when they are getting closer to the object or obstacle [16,17]. The risk potential ($U_{risk}$) is expressed as the sum of the road center potential and road boundary potential. It can be calculated using the following equation:

$$U_{risk}(X,Y) = U_r(X,Y) + U_l(X,Y), \tag{1}$$

where $U_r$ and $U_l$ indicate the values of the road center and boundary potentials, respectively. The road center potential is expressed as

$$U_r(X,Y) = w_r \left\{ 1 - \exp\left( -\frac{(Y_{center} - Y)^2}{2\sigma_r{}^2} \right) \right\}, \tag{2}$$

where $w_r$ denotes the weighting coefficient, $Y_{center}$ indicates the road center, and $\sigma_r$ represents the variance. Conversely, the road boundary potential is expressed as follows:

$$U_l(X,Y) = w_l \left\{ - \exp\left( -\frac{(Y_l - Y)^2}{\sigma_l{}^2} \right) \right\}, \tag{3}$$

where $w_l$ indicates the weighting coefficient, $Y_l$ denotes the road boundary, and $\sigma_l$ represents the variance. In this paper, the weighting coefficients and the variances are set as follows: $w_r = 7.4 \times 10^4$, $w_l = 1.0 \times 10^5$, $\sigma_r = 2.0$, and $\sigma_l = 0.6$.

### 2.2. Optimization of the Reference Yaw Rate

Equations (4–10) show the optimization algorithm for the reference yaw rate. The algorithm is expressed as

$$r^* = r + \Delta r_p{}^* \tag{4}$$

$$\Delta r_p{}^* = \underset{\Delta r_p(i)}{\text{argmin}} J \tag{5}$$

with

$$J = \sum_{j=1}^{N} \left[ U_{risk}\big(X_p(i,j), Y_p(i,j)\big) + r_y \Delta r_p{}^2(i) \right] \tag{6}$$

subject to

$$X_p = X + \int_0^{t_p} (V + a_x t_p) \cos(\psi + r_p t_p) dt_p \tag{7}$$

$$Y_p = Y + \int_0^{t_p} \left(V + a_x t_p\right) \sin\left(\psi + r_p t_p\right) dt_p \tag{8}$$

$$\left|\Delta r_p\right| \leq 0.1 \tag{9}$$

$$\left|V \times r\right| \leq 5.0 \tag{10}$$

The reference yaw rate increment or decrement ($\Delta r_p{}^*$) that minimizes the cost of the evaluation function (Equation (6)) is selected, and the reference yaw rate ($r^*$) is obtained by adding $\Delta r_p{}^*$ to the existing yaw rate ($r$). This implies that the selected yaw rate must minimize the risk potential of the predictive position and achieve smooth vehicle yawing motion. The predicted vehicle positions $X_p$ and $Y_p$ are calculated by Equations (7) and (9). The constraints on the reference yaw rate increment or decrement candidate and the vehicle state (lateral acceleration) for this optimal control problem are Equations (9) and (10). This optimal control problem must be solved periodically to react to unforeseen changes in the environment, such as prediction errors of the dynamic obstacles. This receding horizon concept is generally used in model predictive control, and a solver such as the generalized minimal residual method (GMRES) can be applied [18,19]. Figure 2 depicts the overview of the optimization of the reference yaw rate.

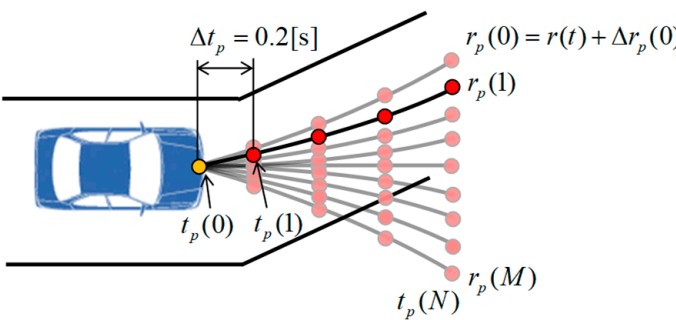

**Figure 2.** Overview of the optimization of the reference yaw rate.

Herein, $i$ denotes the sampling frequency of the prediction time, $j$ indicates the sampling frequency of the reference yaw rate increment or decrement candidate, $N$ represents the prediction time horizon, $M$ denotes the reference yaw rate candidate horizon, $r_y$ indicates the weighting coefficient, $\Delta r_p$ denotes the reference yaw rate increment or decrement candidate, $t_p$ indicates the predicted time, and $r_p$ denotes the reference yaw rate candidate. In this paper, the weighting coefficient is set as follows: $r_y = 70$.

The constraint of the lateral displacement along the course of collision avoidance is a soft one. This may be violated as the optimization is trying to find the minimum risk path under the constrained reference yaw rate increment or decrement (the absolute value is limited to 0.1 rad/s).

### 2.3. Linear Vehicle Model

Figure 3 illustrates the linear equivalent bicycle model of the 4WS employed in this study, with one wheel each for the front and rear to design the feedforward controller.

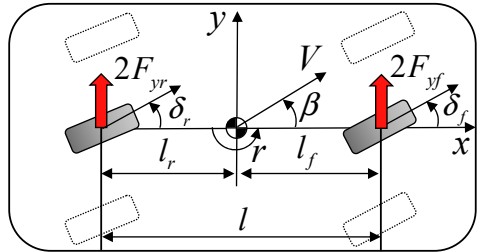

**Figure 3.** Linear equivalent two-degree-of-freedom (2-DOF) bicycle model of the 4WS.

Equations (11) and (12) indicate the lateral and yaw motions, respectively:

$$mV\left(\dot{\beta}+r\right) = 2F_{yf} + 2F_{yr}, \tag{11}$$

$$I_z\dot{r} = 2F_{yf}l_f - 2F_{yr}l_r, \tag{12}$$

where $m$ denotes the vehicle mass, $F_{xf}$ and $F_{yr}$ indicate the front and rear cornering forces, respectively, $I_z$ represents the yaw moment of inertia, and $l_f$ and $l_r$ denote the distances from the front and rear axles to the center of gravity, respectively. If the lateral force generated by the tire is proportional to the tire sideslip angle, the equations for the lateral and yaw motions can be expressed as follows:

$$mV\left(\dot{\beta}+r\right) = 2C_f\left(\delta_f - \frac{l_f}{V}r - \beta\right) + 2C_r\left(\delta_r + \frac{l_r}{V}r - \beta\right), \tag{13}$$

$$I_z\dot{r} = 2C_fl_f\left(\delta_f - \frac{l_f}{V}r - \beta\right) - 2C_rl_r\left(\delta_r + \frac{l_r}{V}r - \beta\right), \tag{14}$$

where $C_f$ and $C_r$ denote the front and rear cornering stiffness, respectively.

### 2.4. Feedforward Controller

Equations (13) and (14) are Laplace transformed and solved for the rear steering angle ($\delta_r$), assuming a steady state. Therefore, the feedforward term of the desired rear steering angle ($\delta_r{}^*{}_{ff}$) can be defined as

$$\delta_r{}^*{}_{ff} = \delta_f + \left\{\frac{mV\left(l_fC_f - l_rC_r\right)}{2C_fC_rl} - \frac{l}{V}\right\}r^*. \tag{15}$$

This equation indicates the rear steering angle required to generate the reference yaw rate ($r^*$) when the driver is turning at a specific front steering angle ($\delta_f$).

### 2.5. Feedback Controller

In addition to the feedforward controller, a feedback controller is used to compensate for the influence of external disturbances or dynamic uncertainties [8]. The optimal linear quadratic regulator (LQR) is designed to reduce the error between the actual vehicle motion and the reference value to zero.

If the error variables from the desired vehicle sideslip angle ($\beta^*$), which is $\beta^* = 0$ rad in this study, and the reference yaw rate ($r^*$) are $\Delta\beta$ and $\Delta r$, respectively, they can be expressed as follows:

$$\Delta\beta = \beta - \beta^*, \tag{16}$$

$$\Delta r = r - r^*. \tag{17}$$

Therefore, the error state equation can be obtained from Equations (13) and (14) as

$$\begin{bmatrix} \Delta\dot{\beta} \\ \Delta\dot{r} \end{bmatrix} = \begin{bmatrix} A_{11} & A_{12} \\ A_{21} & A_{22} \end{bmatrix}\begin{bmatrix} \Delta\beta \\ \Delta r \end{bmatrix} + \begin{bmatrix} B_1 \\ B_2 \end{bmatrix}\delta^*_{r\_fb}, \tag{18}$$

where all elements in the matrix are determined from the equations for two-degree-of-freedom (2-DOF) motion as follows:

$$A_{11} = -\frac{2\left(C_f+C_r\right)}{mV}, \ A_{12} = -1 - \frac{2\left(l_fC_f - l_rC_r\right)}{mV^2}, \ B_1 = \frac{2C_r}{mV}$$

$$A_{21} = -\frac{2\left(l_fC_f - l_rC_r\right)}{I_z}, \ A_{22} = -\frac{2\left(l_f^2C_f + l_r^2C_r\right)}{I_zV}, \ B_2 = -\frac{2l_rC_r}{I_z}$$

Furthermore, the feedback term of the desired rear-wheel steering angle ($\delta_r^*{}_{\_fb}$) can be defined by applying the error state feedback to the error state equation as follows:

$$\delta_r^*{}_{\_fb} = -K_\beta \Delta\beta - K_r \Delta r, \tag{19}$$

where $K_\beta$ and $K_r$ indicate the state feedback gains. The LQ control theory is used to design the regulator, and the state feedback gain is determined to minimize this evaluation function.

$$J_{LQ} = \int_0^\infty \left[ \left( \frac{\Delta\beta}{\beta_{max}} \right)^2 + \left( \frac{\Delta r}{r_{max}} \right)^2 + \left( \frac{\delta_{r\_fb}^*}{\delta_{rmax}} \right)^2 \right] dt, \tag{20}$$

where the denominator of each term in the evaluation function indicates the tolerance of the error of the state quantity and the limit of the control input. In this paper, the control parameters are set as follows: $\beta_{max} = 0.1$ rad, $r_{max} = 0.1$ rad/s, and $\delta_r = 0.1$ rad. The stability of the model matching the control of the active rear wheel steering control is guaranteed, as the weighting coefficients of the performance index are set in the way that the poles of the feedback-controlled vehicle are located on the left side of the complex plane. Regarding the system stability, the closed loop poles can be determined by calculating the eigenvalues of the closed loop matrix $A - BK$:

$$\dot{x} = Ax + Bu = (A - BK)x, \tag{21}$$

where the feedback gain matrix $K$ is calculated by the LQ control theory to minimize the cost function:

$$J_{LQ} = \int_0^\infty x^T Q x + u^T R u \, dt, \tag{22}$$

The Riccati equation for finding the feedback gain is expressed as follows:

$$PA + A^T P - PBR^{-1}B^T P + Q = 0, \tag{23}$$

Using $P$, which satisfies Equation (23), the feedback gain is calculated as follows:

$$K = R^{-1}B^T P. \tag{24}$$

Here, all elements of each matrix are represented as follows:

$$A = \begin{bmatrix} A_{11} & A_{12} \\ A_{21} & A_{22} \end{bmatrix}, B = \begin{bmatrix} B_1 \\ B_2 \end{bmatrix}, x = \begin{bmatrix} \Delta\beta \\ \Delta r \end{bmatrix}, u = \begin{bmatrix} \delta_{r\_fb}^* \end{bmatrix}$$

$$K = \begin{bmatrix} K_\beta & K_r \end{bmatrix}, Q = \begin{bmatrix} \frac{1}{\beta_{max}^2} & 0 \\ 0 & \frac{1}{r_{max}^2} \end{bmatrix}, R = \begin{bmatrix} \frac{1}{\delta_{rmax}^2} \end{bmatrix}$$

Moreover, in the control part, the constraint of the control input (rear-wheel steering) is under a hard constraint ($\delta_r \leq 3.0$ deg). In addition, for securing the vehicle stability during collision avoidance, as the physical limit of the vehicle motion, each tire force should not exceed its friction limit as shown in the following expression:

$$\sqrt{F_x^2 + F_y^2} \leq \mu F_z, \tag{25}$$

where $F_x$ and $F_y$ indicate the tire longitudinal and lateral forces, respectively, $\mu$ denotes the road friction coefficient, and $F_z$ indicates the tire vertical load.

The closed loop system poles with respect to the change of the velocity can be calculated as the following complex plane (Figure 4).

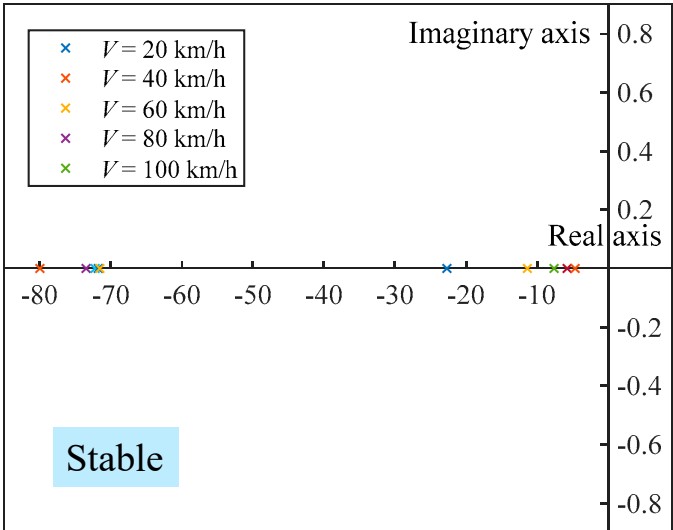

**Figure 4.** Closed loop system poles $(A-BK)$ with respect to the change of the velocity.

## 3. Model Effectiveness Verification Using a Simulation

In this section, the rear-wheel steering control proposed in Section 2 is applied to a double lane change test assuming emergency avoidance, and its effectiveness is verified through a simulation. Furthermore, the nonlinear four-wheel vehicle model used in the simulation, reference driver model, and zero-sideslip angle 4WS are explained in detail.

### 3.1. Nonlinear Four-Wheel Vehicle Model

We used a 4-DOF vehicle model, considering the longitudinal, lateral, yawing, and rolling motions in this simulation. However, the vertical and pitching motions were neglected to simplify the model. The equations of the longitudinal, lateral, yawing, and rolling motions can be expressed as follows:

$$m(\dot{u} - vr) = \left(F_{xfl} + F_{xfr}\right)\cos\delta_f - \left(F_{yfl} + F_{yfr}\right)\sin\delta_f + (F_{xrl} + F_{xrr})\cos\delta_r - \left(F_{yrl} + F_{yrr}\right)\sin\delta_r, \tag{26}$$

$$m(\dot{v} + ur) - m_s h_s \ddot{\varphi} = \left(F_{xfl} + F_{xfr}\right)\sin\delta_f + \left(F_{yfl} + F_{yfr}\right)\cos\delta_f + (F_{xrl} + F_{xrr})\sin\delta_r + \left(F_{yrl} + F_{yrr}\right)\cos\delta_r \tag{27}$$

$$\begin{aligned} I_z \dot{r} - I_{xz}\ddot{f} = & \; l_f\left\{\left(F_{xfl} + F_{xfr}\right)\sin\delta_f + \left(F_{yfl} + F_{yfr}\right)\cos\delta_f\right\} - l_r\left\{(F_{xrl} + F_{xrr})\sin\delta_r + \left(F_{yrl} + F_{yrr}\right)\cos\delta_r\right\} \\ & + \tfrac{d}{2}\left\{\left(F_{xfr} - F_{xfl}\right)\cos\delta_f + (F_{xrr} - F_{xrl})\cos\delta_r + \left(F_{yfl} - F_{yfr}\right)\sin\delta_f + \left(F_{yrl} - F_{yrr}\right)\sin\delta_r\right\} \end{aligned} \tag{28}$$

$$I_\varphi \ddot{\varphi} - I_{xz}\dot{r} = h_s m_s(\dot{v} + ur) + h_s m_s g \sin\varphi - \left(K_\varphi \varphi + C_\varphi \dot{\varphi}\right), \tag{29}$$

where $F_{xij}$ and $F_{yij}$ indicate the tire longitudinal and lateral forces, respectively, subscript $i$ denotes the front ($f$) and rear ($r$) wheels, subscript $j$ denotes the left ($l$) and right ($r$) wheels, $u$ and $v$ represent the longitudinal and lateral velocities, respectively, $m_s$ denotes the sprung mass, $d$ indicates the tread width, $I_\varphi$ and $I_{xz}$ denote the roll moment and yaw tensor of inertia, respectively, $\varphi$ indicates the roll angle, $h_s$ represents the roll moment arm, $K_\varphi$ denotes the roll stiffness, and $C_\varphi$ indicates the roll viscous damping coefficient.

### 3.2. Reference Driver Model

To verify the effectiveness of the proposed control method, we developed a reference driver model that steered the front-wheel steering vehicle (2WS). Based on the linear equivalent bicycle model of the 2WS vehicle, the reference front steering angle that realized the reference yaw rate obtained in Section 2.2 could be determined in a feedforward manner as follows:

$$\delta_f{}^* = \frac{(1 + A_s V^2)l}{V} r^*, \tag{30}$$

where $\delta_f^*$ indicates the reference front steering angle and $A_s$ denotes the stability factor. The vehicle behavior observed when the front wheels were steered according to Equation (19) is the ideal behavior for safe and smooth driving.

### 3.3. Zero-Sideslip Angle 4WS

In this section, the conventional zero-sideslip angle rear-wheel steering control method is compared with the proposed rear-wheel steering system.

When the rear wheel is steered with respect to the front wheel in the relationship indicated by a certain transfer function, the rear steering angle is expressed as

$$\delta_r(s) = k(s)\,\delta_f(s), \tag{31}$$

where $k(s)$ indicates the transfer function of the rear steering angle to the front steering angle. The vehicle sideslip angle relative to the front steering angle is expressed as follows:

$$\frac{\beta(s)}{\delta_f(s)} = \frac{\begin{vmatrix} 2\left\{C_f + k(s)C_r\right\} & mV + \frac{2\left(l_f C_f - l_r C_r\right)}{V} \\ 2\left\{l_f C_f - k(s)l_r C_r\right\} & I_z s + \frac{2\left(l_f{}^2 C_f + l_r{}^2 C_r\right)}{V} \end{vmatrix}}{\begin{vmatrix} mVs + 2\left(C_f + C_r\right) & mV + \frac{2\left(l_f C_f - l_r C_r\right)}{V} \\ 2\left(l_f C_f - l_r C_r\right) & I_z s + \frac{2\left(l_f{}^2 C_f + l_r{}^2 C_r\right)}{V} \end{vmatrix}}. \tag{32}$$

Assuming that the numerator in Equation (32) = 0, $k(s)$ can be obtained as follows:

$$k(s) = -\frac{l_r - \frac{ml_f}{2lC_r}V^2 + \frac{I_z V}{2lC_r}s}{l_f + \frac{ml_r}{2lC_f}V^2 + \frac{I_z V}{2lC_f}s} = \frac{k_0}{1 + T_e s} - \frac{C_f}{C_r}\frac{T_e s}{1 + T_e s}, \tag{33}$$

where $k_0$ and $T_e$ are expressed as follows:

$$k_0 = -\frac{l_r\left(1 - \frac{ml_f}{2ll_r C_r}V^2\right)}{l_f\left(1 + \frac{ml_r}{2ll_f C_f}V^2\right)}, \tag{34}$$

$$T_e = \frac{I_z V}{2ll_f C_f + ml_r V^2}. \tag{35}$$

Therefore, if the rear wheels are steered with respect to the front wheels using a transfer function, such as Equation (33), a vehicle with the sideslip angle always set to zero can be realized [20].

### 3.4. Simulation Conditions

The double lane change test considering an emergency avoidance scene was used to verify the effectiveness of the proposed rear-wheel steering system with a human driver model in the loop. Figure 5 depicts the target course developed based on the curved line indicated in Equation (36) [21].

$$y = \begin{cases} \frac{3.5}{2}\left[1 + \tanh\left\{\frac{2}{30}\pi\left(x - X_1 - \frac{30}{2}\right)\right\}\right] & (X_1 < x \leq X_2) \\ \frac{3.5}{2}\left[1 - \tanh\left\{\frac{2}{25}\pi\left(x - X_3 - \frac{25}{2}\right)\right\}\right] & (X_3 < x \leq X_4) \end{cases}, \tag{36}$$

where $X_1$ and $X_3$ indicate the lane change start points and $X_2$ and $X_4$ represent the lane change endpoints. Based on the risk potential described in Section 2.1 and the target course depicted in Figure 5, the risk potential field was generated as illustrated in Figure 6.

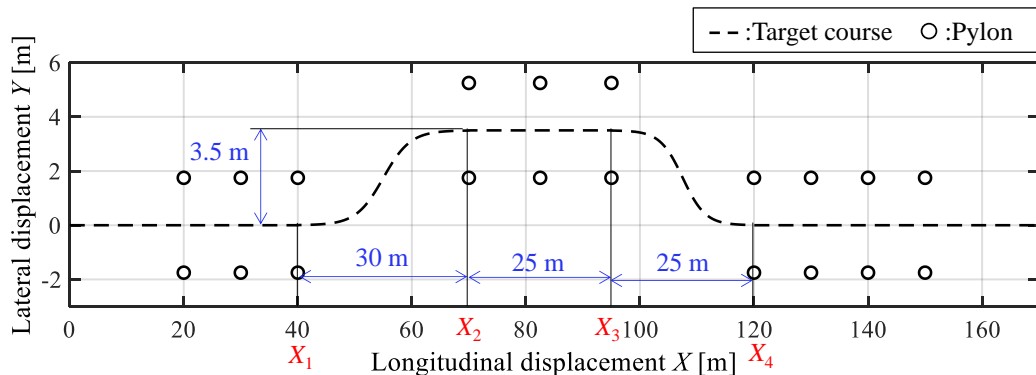

**Figure 5.** Target course of the simulated double lane change test.

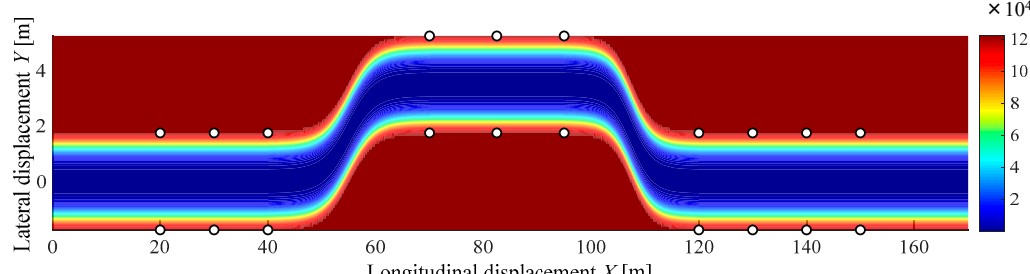

**Figure 6.** Risk potential field of the target course.

The simulations were conducted under the following four conditions with different types of control and with or without control, wherein the vehicle velocity was maintained at a constant $V = 60$ km/h:

1. 2WS based on the reference driver model;
2. 2WS without control;
3. Zero-sideslip angle 4WS;
4. Risk potential field 4WS (the proposed system).

However, the front steering angle inputs for conditions 2, 3, and 4 were determined using a first-order preview-predictive driver model. The driver model for lane-tracking control can be expressed as follows:

$$\delta_{sw} = \frac{h_d}{T_r s + 1} \left[ y_{OL} - (y_c + T_p V \psi) \right], \tag{37}$$

where $\delta_{sw}$ denotes the steering wheel angle, $h_d$ indicates the driver corrective steering gain, $T_r$ represents the driver model steering delay time constant, $y_{OL}$ denotes the desired preview lateral displacement, $y_c$ indicates the existing lateral displacement of the vehicle, $T_p$ represents the driver model predictive time, and $\psi$ denotes the existing yaw angle of the vehicle with respect to the desired lane. Figure 7 illustrates an overview of the first-order preview-predictive driver steering model. In the simulation described in the next subsection, the values of driver parameters were set as follows: $h_d = 0.4$ rad/m, $T_p = 1.3$ s, and $T_r = 0.2$ s.

### 3.5. Simulation Results

The time histories of the front steering angle, rear steering angle, yaw rate, and vehicle sideslip angle are depicted in Figure 8. Figures 9 and 10 illustrate the vehicle trajectory and Lissajous diagram of the steering wheel angle and yaw rate, respectively. Figure 11 depicts the integrated risk potential values for the trajectory of the vehicle over the risk potential field.

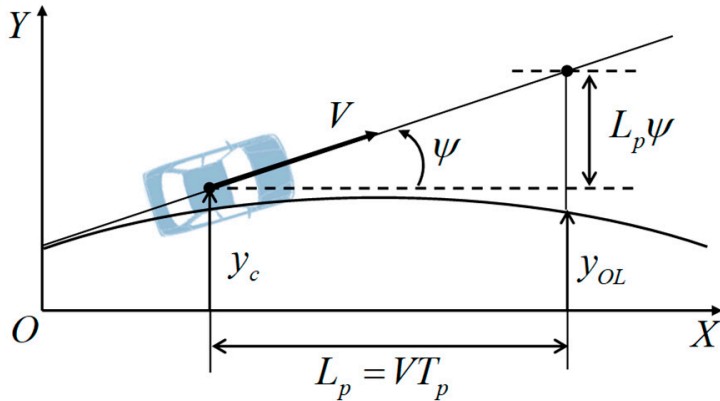

**Figure 7.** First-order preview-predictive driver steering model.

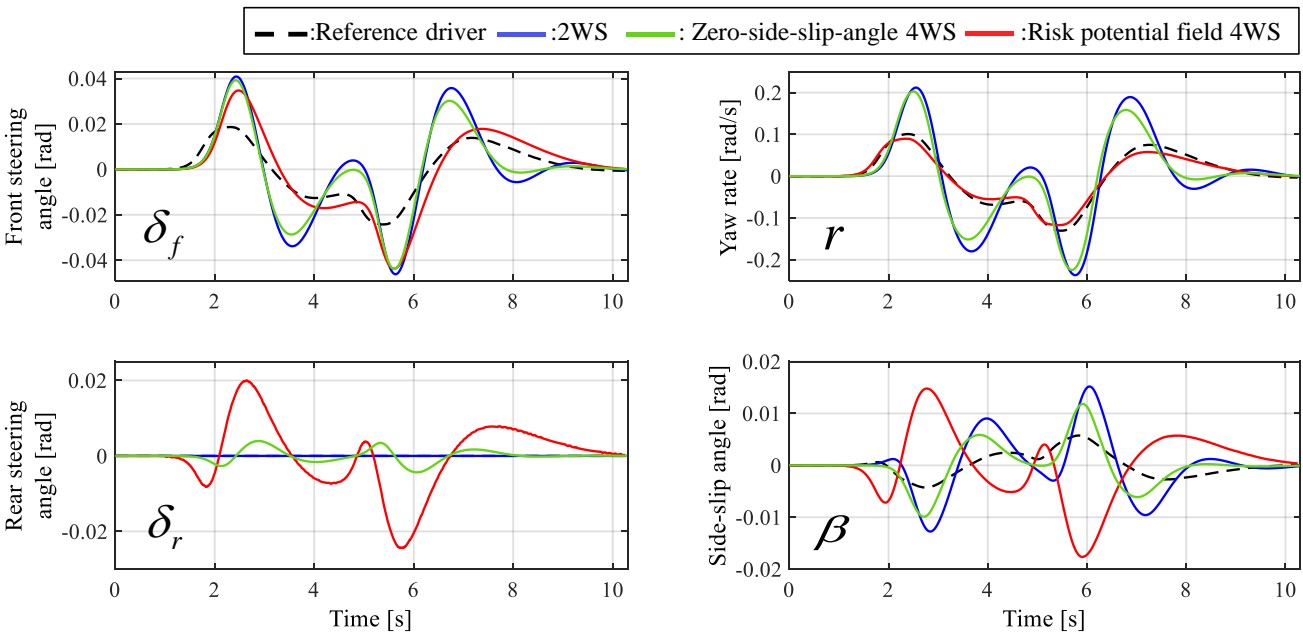

**Figure 8.** Comparison of vehicle behaviors.

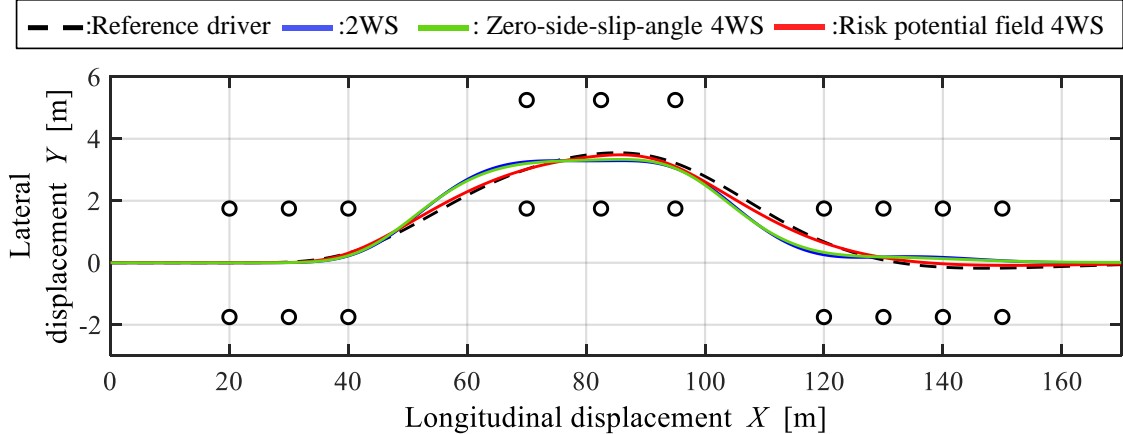

**Figure 9.** Comparison of vehicle trajectories.

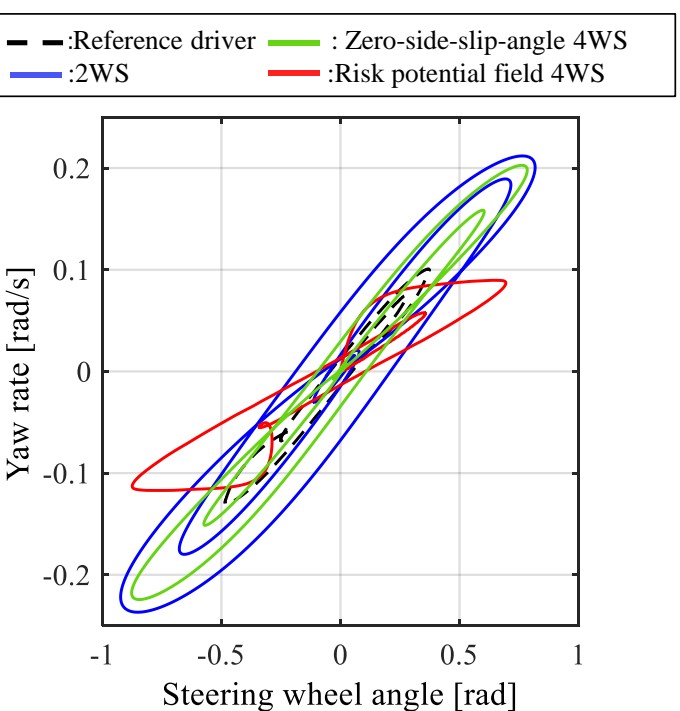

**Figure 10.** Comparison of Lissajous diagrams of the steering wheel angle and yaw rate.

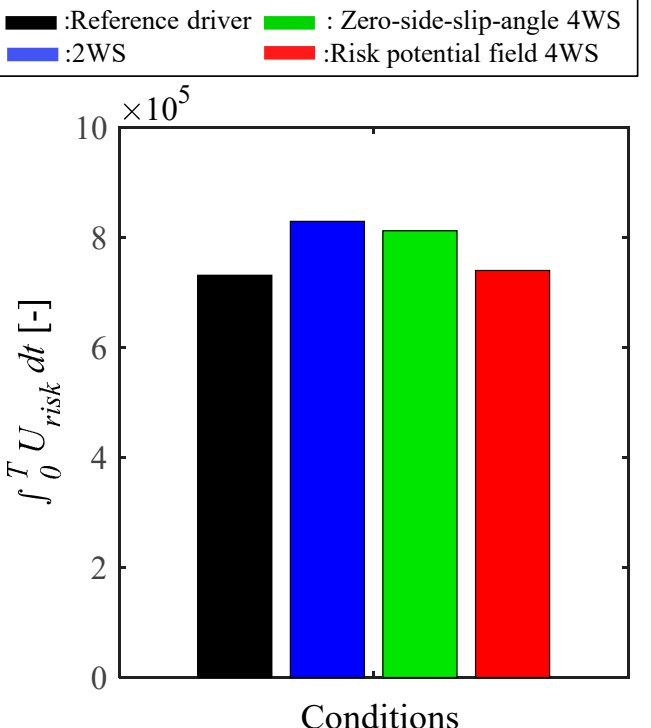

**Figure 11.** Integrated risk potential values.

Figure 8 indicates that when the risk potential field 4WS was applied, the rear wheels were actively steered, and the corrective steering of the front wheels reduced in comparison with that observed in the cases without control and with a zero-sideslip angle 4WS. Additionally, we observed that the yaw rate was equivalent to that of the reference driver, owing to the application of the risk potential field 4WS. Conversely, although the sideslip angle tended to increase when the risk potential field 4WS was applied, it attained

a value of approximately 0.02 rad, which was within the acceptable range, considering the vehicle stability. Figure 9 indicates that when the risk potential field 4WS was applied, the obtained trajectory was similar to that of the reference driver. Consequently, safe and smooth driving was achieved. In the cases of a 2WS without control and a zero-sideslip angle 4WS (Figure 10), a larger steering wheel angle was required to follow the double lane change course. Furthermore, in the cases of the 2WS without control and zero-sideslip angle 4WS, several large whirls were observed in the Lissajous diagram, which indicates that the driver needed to implement corrective steering extensively. This implies that it was difficult for the driver to steer the vehicle to follow the course. However, the whirls were smaller and fewer in the case of the risk potential field 4WS, indicating that that the vehicle was easier to steer for following the course. Finally, Figure 11 depicts that by applying the risk potential field 4WS, the vehicle could drive in a low-risk area and reduce the risk to the same level as the reference driver.

## 4. Experimental Study Using a Driving Simulator

This section describes the experimental study of the proposed rear-wheel steering system implemented in a double lane change test using a driving simulator with a four-axis motion cueing system.

### 4.1. Experimental Conditions

Figure 12 illustrates the double lane change scenario used for conducting the experiment, assuming an emergency avoidance scene. The experiments were performed considering two scenarios with constant vehicle velocities of $V = 60$ km/h and 80 km/h. Ten drivers (S1–S10) were employed to drive through the driving course for each of the two scenarios under the following conditions:

1. 2WS without control;
2. Zero-sideslip angle 4WS;
3. Risk potential field 4WS (the proposed system).

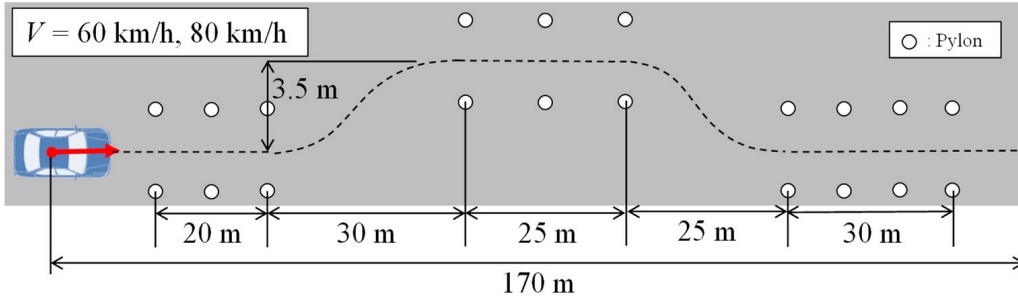

**Figure 12.** Course of the double lane change test used for experiments.

The driving test was conducted 3 times for each condition, and a total of 18 experimental data points were obtained per person. However, the drivers input only the steering wheel angle and not the acceleration or deceleration. The effectiveness of the proposed method was examined by comparing the data obtained in these experiments with those of the vehicle behavior, based on the reference driver model described in Section 3.2.

IPG CarMaker 8.0.1® software was used to visualize the vehicle motion simulation. We used a complete vehicle model that considered the longitudinal, lateral, yawing, rolling, pitching, and vertical motions. Additionally, the driving simulator with a four-axis motion cueing system was used to allow the drivers to feel the vehicle's behavior. Figure 13 depicts the overview of the experimental equipment. Herein, the driving simulator comprised a reclining driver seat, three pedals (gas, brake, and clutch), a shifting lever, steering wheel with a servo, large and wide display, sound system, computer for dynamics calculation, and four motion electric actuators. The four motion actuators aided in realizing the lateral

and longitudinal acceleration, roll and pitch of the vehicle, and small vibrations from the road surface during the simulation.

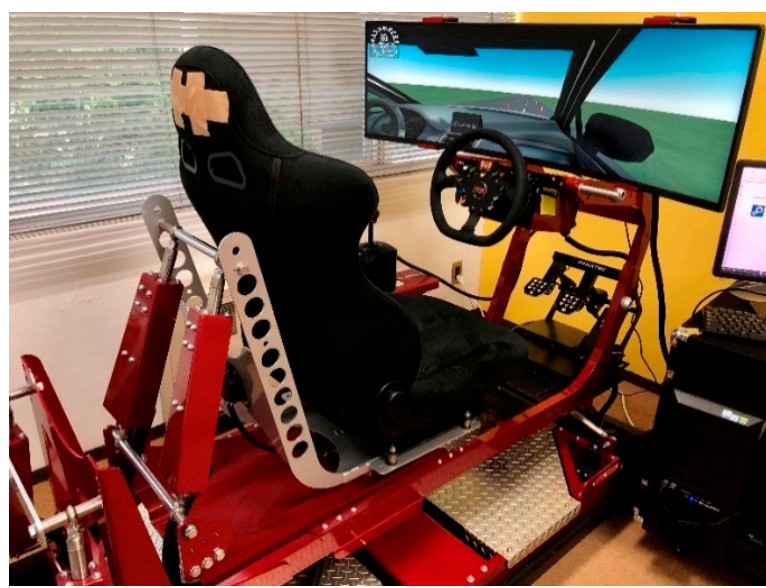

**Figure 13.** Overview of the experimental equipment.

### 4.2. Results

Herein, the results of the experiment with a frequent driver (subject S1) and an occasional driver (subject S6) are presented as representative examples. Figure 14a,b illustrates the time histories of the front steering angle, rear steering angle, yaw rate, and sideslip angle when subject S1 drove three times at 60 km/h and 80 km/h, respectively, under each condition. Figure 15a,b shows the trajectories of subject S1 driving at 60 km/h and 80 km/h, respectively. Figure 16a,b depicts the Lissajous diagrams of the steering wheel angle and yaw rate when subject S1 drove at 60 km/h and 80 km/h, respectively, using the data obtained during the third drive in each condition. Similarly, the respective experimental results of subject S6 are illustrated in Figures 17–19.

We observed that the application of the risk potential field 4WS actively steered the rear wheels in order to make the vehicle follow the reference yaw rate. Additionally, the trajectory was equivalent to that of the reference driver. In the case of the 2WS and zero-sideslip angle 4WS, the yaw rate and trajectory exhibited considerable variations. However, in the case of the risk potential field 4WS, nearly no variation was observed in the results of the three experiments for the yaw rate and trajectory. The Lissajous diagrams indicate that the amount of corrective steering was reduced because smaller and fewer whirls around the origin were generated by the proposed method. Furthermore, the comparison of vehicle trajectories in Figure 18a,b indicates that the difference from the reference driver's trajectory was larger than that observed in the proposed method. Moreover, the variation was higher when the vehicle velocity was 80 km/h than when it was 60 km/h. Therefore, the proposed rear-wheel steering control system provided a more significant control effect at higher velocity regions.

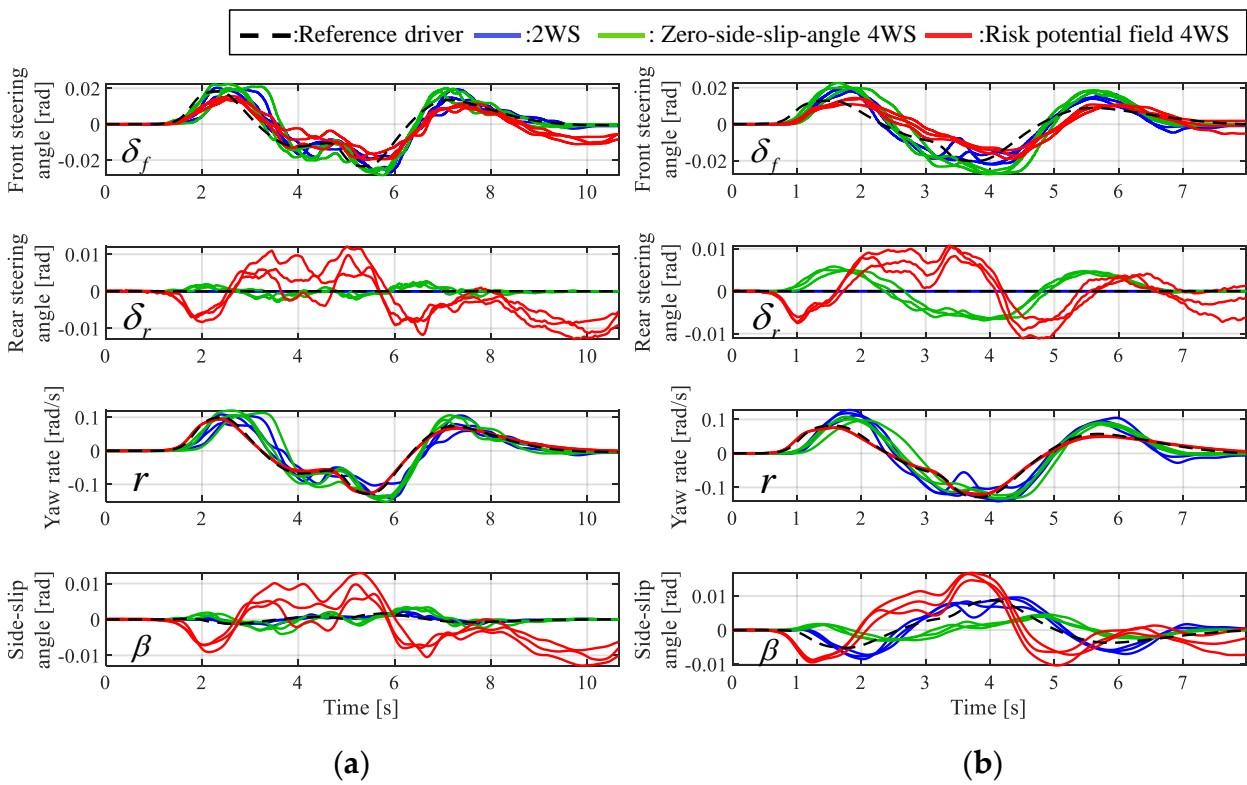

**Figure 14.** Comparison of the vehicle behavior for subject S1 at (**a**) *V* = 60 km/h and (**b**) *V* = 80 km/h.

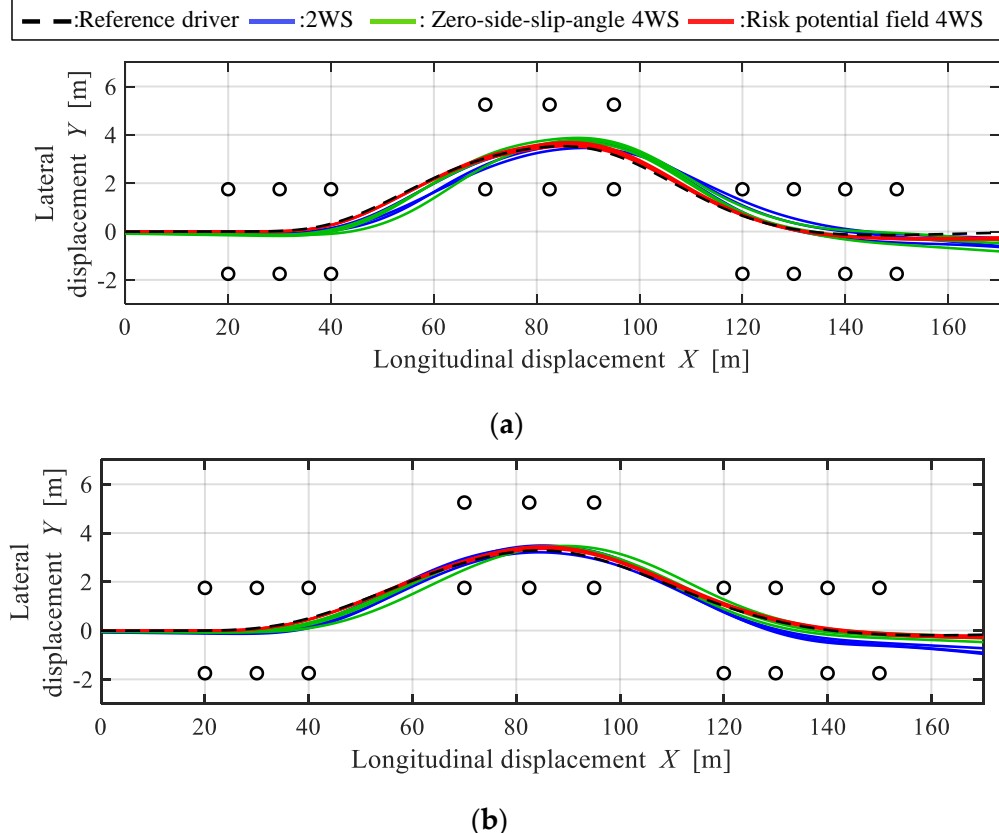

**Figure 15.** Comparison of the vehicle trajectory for subject S1 at (**a**) *V* = 60 km/h and (**b**) *V* = 80 km/h.

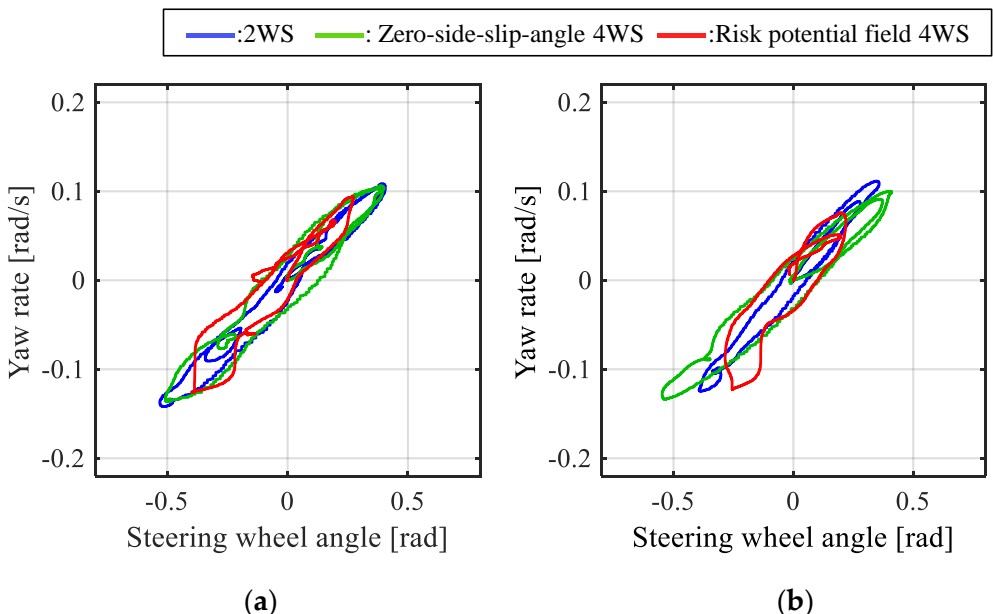

**Figure 16.** Comparison of Lissajous diagrams of the steering angle and yaw rate for subject S1 using the third trial data in each condition at (**a**) *V* = 60 km/h and (**b**) *V* = 80 km/h.

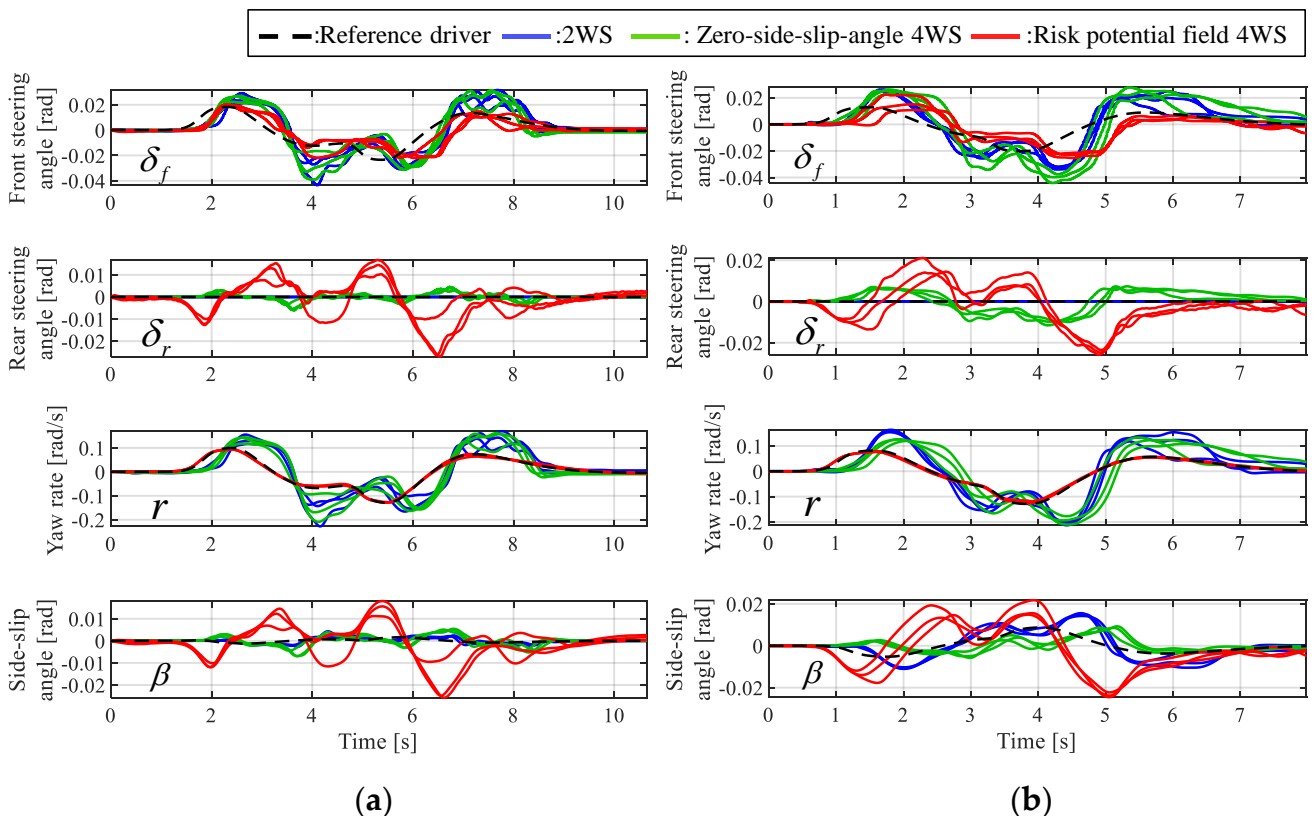

**Figure 17.** Comparison of the vehicle behavior for subject S6 at (**a**) *V* = 60 km/h and (**b**) *V* = 80 km/h.

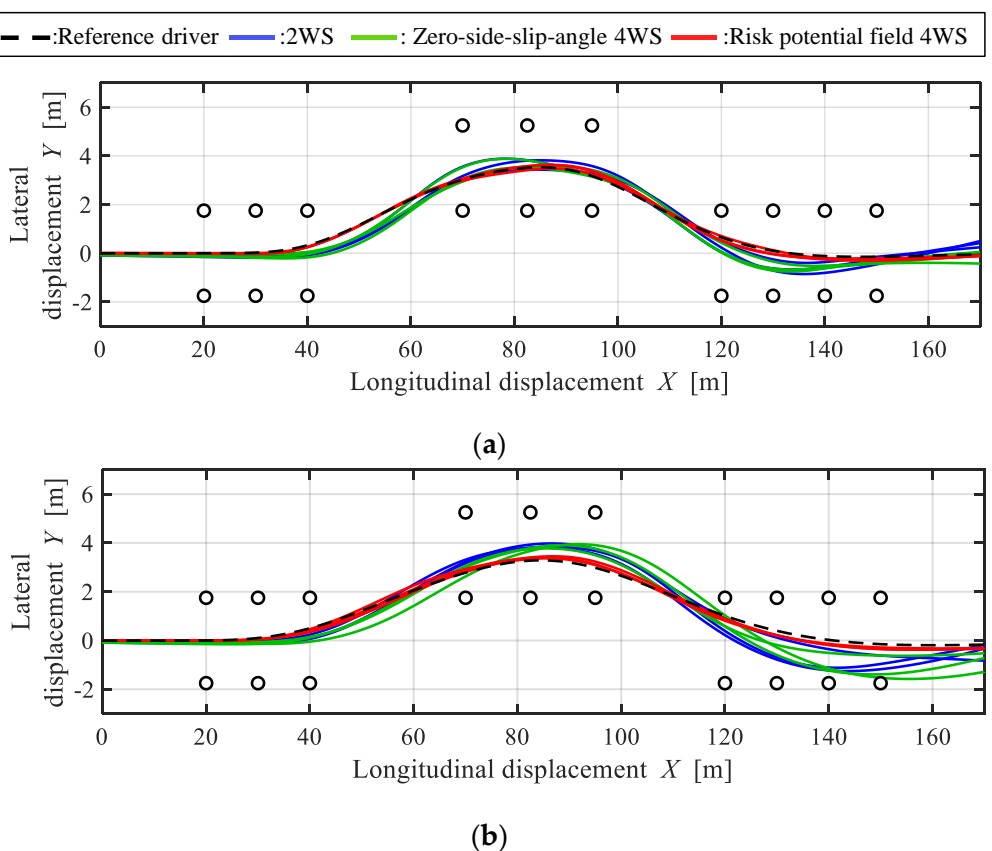

**Figure 18.** Comparison of the vehicle trajectory for subject S6 at (**a**) $V = 60$ km/h and (**b**) $V = 80$ km/h.

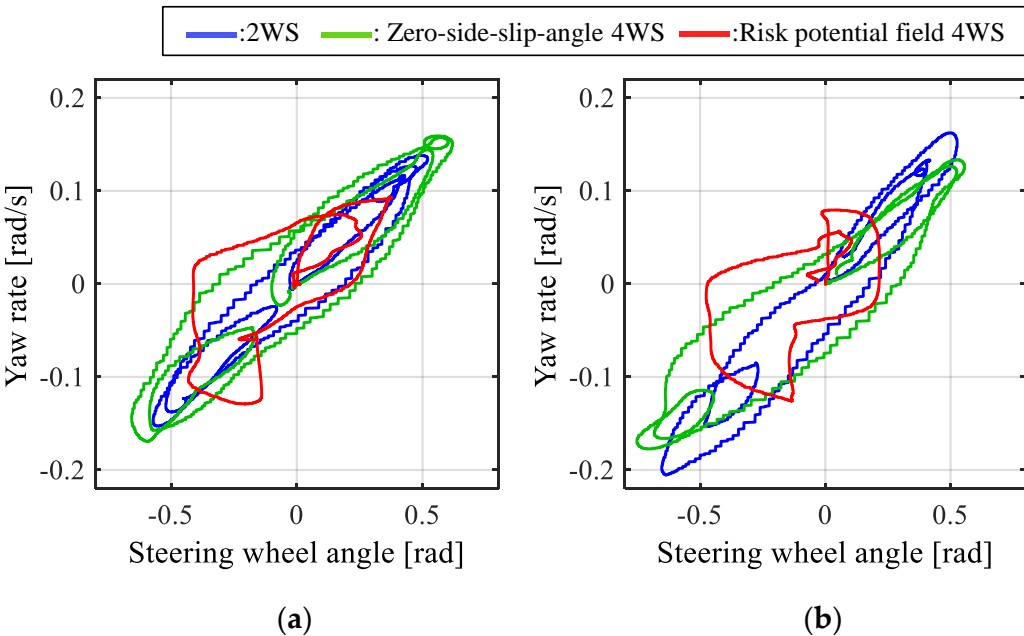

**Figure 19.** Comparison of Lissajous diagrams of the steering angle and yaw rate for subject S6 using the third trial data in each condition at (**a**) $V = 60$ km/h and (**b**) $V = 80$ km/h.

### 4.3. Evaluation Index

To quantitatively evaluate the obtained experimental results (Section 4.2), we used the root mean square (RMS) value of the lateral deviation ($y_{cr}$), integral of the squared steering wheel angle, the Emergency Avoidance Performance Index (EAPI), and the RMS value of

the front steering angle deviation ($\delta_{fcr}$) for all subjects. Herein, the lateral deviation, EAPI, and front steering angle deviation can be defined as follows:

- Lateral deviation ($y_{cr}$)

The lateral deviation is the difference between the $y$-coordinates of the trajectories when driven by the reference driver model and the subject in the experiment. This can be expressed as

$$y_{cr} = y_{ref} - y, \tag{38}$$

where $y_{ref}$ denotes the $y$-coordinate of the trajectory when driven by the reference driver model and $y$ indicates the $y$-coordinate of the trajectory obtained in the experiment with the subject. The smaller the absolute value of this lateral deviation, the closer the driving of the subject is to that of the reference driver, ensuring a safer and smoother drive.

- EAPI [22]

The EAPI is determined as the area integrated along the curve of the steering wheel angle with respect to the yaw rate. The smaller the amount of corrective steering, the smaller the index, indicating better handling quality. This index can be defined as follows:

$$S = \frac{1}{2} \int_0^T \left( \delta_{sw}^2 + r^2 \right) d\left( \tan^{-1}\left( \frac{r}{\delta_{sw}} \right) \right) = \frac{1}{2} \int_0^T \left( \delta_{sw}\dot{r} - \dot{\delta}_{sw}r \right) dt. \tag{39}$$

- Front steering angle deviation ($\delta_{fcr}$)

The front steering angle deviation is the difference between the front steering angles when driven by the reference driver model and the subject in the experiment. This can be expressed as

$$\delta_{fcr} = \delta_{fref} - \delta_f, \tag{40}$$

where $\delta_{fref}$ denotes the front steering angle when driven by the reference driver model and $\delta_f$ indicates the front steering angle obtained in the experiment with the subject. The smaller the absolute value of the front steering angle deviation, the closer the driving of the subject is to that of the reference driver.

### 4.4. Discussion

Figure 20a,b depicts the RMS values of the lateral deviations for each subject and the mean of all the data. However, the data for each subject exhibited an average of three times for each condition. Similarly, Figure 21a,b illustrates the integral of the squared steering wheel angle, while Figure 22a,b presents the EAPI. Figure 23 depicts the integrated risk potential values.

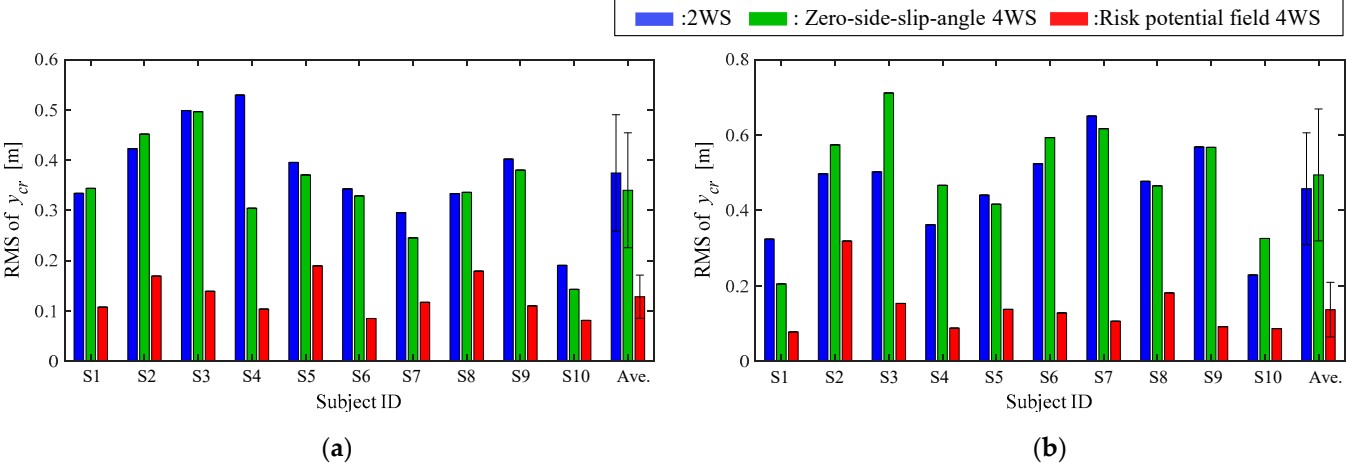

(**a**)  (**b**)

**Figure 20.** Root mean square (RMS) of the lateral deviation at (**a**) $V = 60$ km/h and (**b**) $V = 80$ km/h.

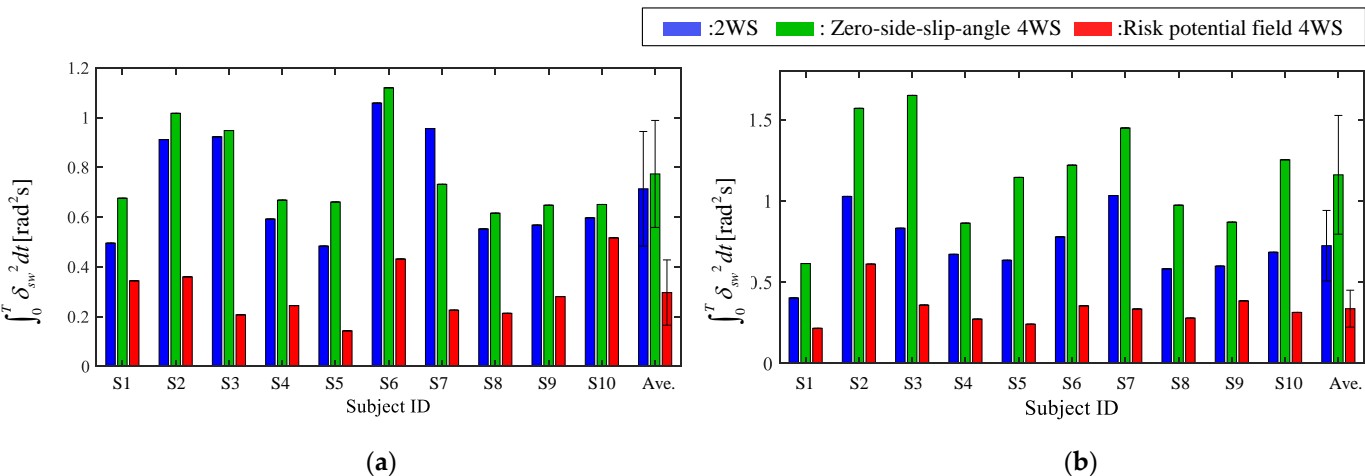

**Figure 21.** Integral of the squared steering wheel angle at (**a**) $V = 60$ km/h and (**b**) $V = 80$ km/h.

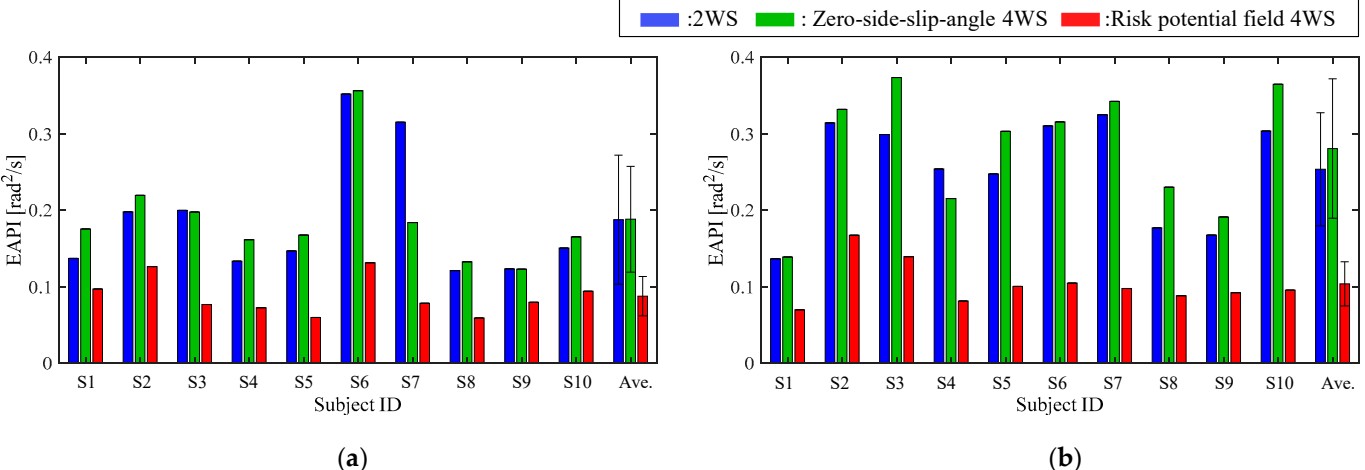

**Figure 22.** Emergency avoidance performance index (EAPI) at (**a**) $V = 60$ km/h and (**b**) $V = 80$ km/h.

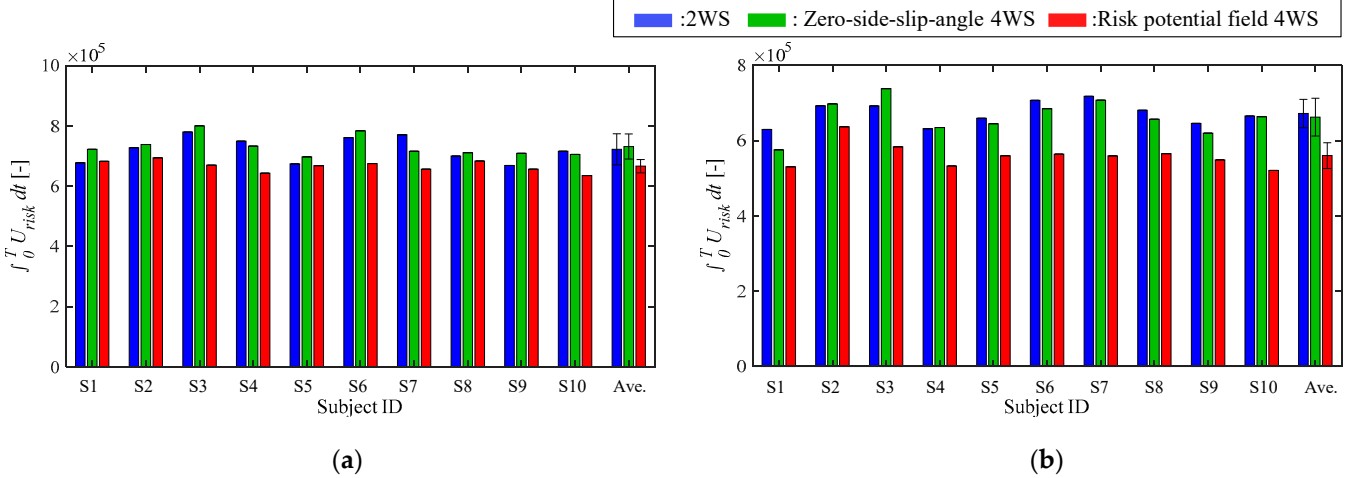

**Figure 23.** Integrated risk potential values at (**a**) $V = 60$ km/h and (**b**) $V = 80$ km/h.

Figure 20 indicates that the application of the risk potential field 4WS reduced the RMS value of the lateral deviation significantly for all subjects in comparison with the cases of the 2WS and zero-sideslip angle 4WS. Figure 21 verifies that when the risk potential field

4WS was applied, the steering burden of the driver reduced for all subjects compared with the cases of the 2WS and zero-sideslip angle 4WS. Figure 22 validates that the application of the risk potential field 4WS reduced the EAPI significantly for all subjects in comparison with the cases of the 2WS and zero-sideslip angle 4WS. Furthermore, the comparison of Figure 22a,b indicates that the decrease in EAPI when the risk potential field 4WS was applied was larger when the vehicle velocity was 80 km/h than that observed for 60 km/h. In other words, the higher the vehicle velocity, the more difficult it is to conduct emergency avoidance without control. However, the emergency avoidance performance was significantly improved, owing to the application of the proposed method. Finally, Figure 23 indicates that the risk potential of the vehicle trajectory could be reduced by using the risk potential field 4WS.

Figure 24 illustrates the correlation diagrams between the RMS of the front steering angle deviation and that of the rear steering angle intervened by the control system for all subjects. However, each RMS was the average of using the risk potential field 4WS three times. The correlation coefficients in Figure 24a,b are 0.9954 and 0.9981, respectively, both of which exhibit strong positive correlations. Figure 24 indicates that the larger the front steering angle deviation, the larger the rear steering angle. In other words, the amount of support for the rear wheels was larger for drivers who deviated from the steering behaviors of the reference driver, whereas the amount of support for the rear wheels was smaller for drivers who closely followed the steering behaviors of the reference driver. Therefore, the system could guide drivers with minimal driving experience safely through the course through active assistance. Moreover, by minimizing the assistance intervention for experienced drivers, the system could guide them safely through the course while maintaining the driver's vehicle control authority.

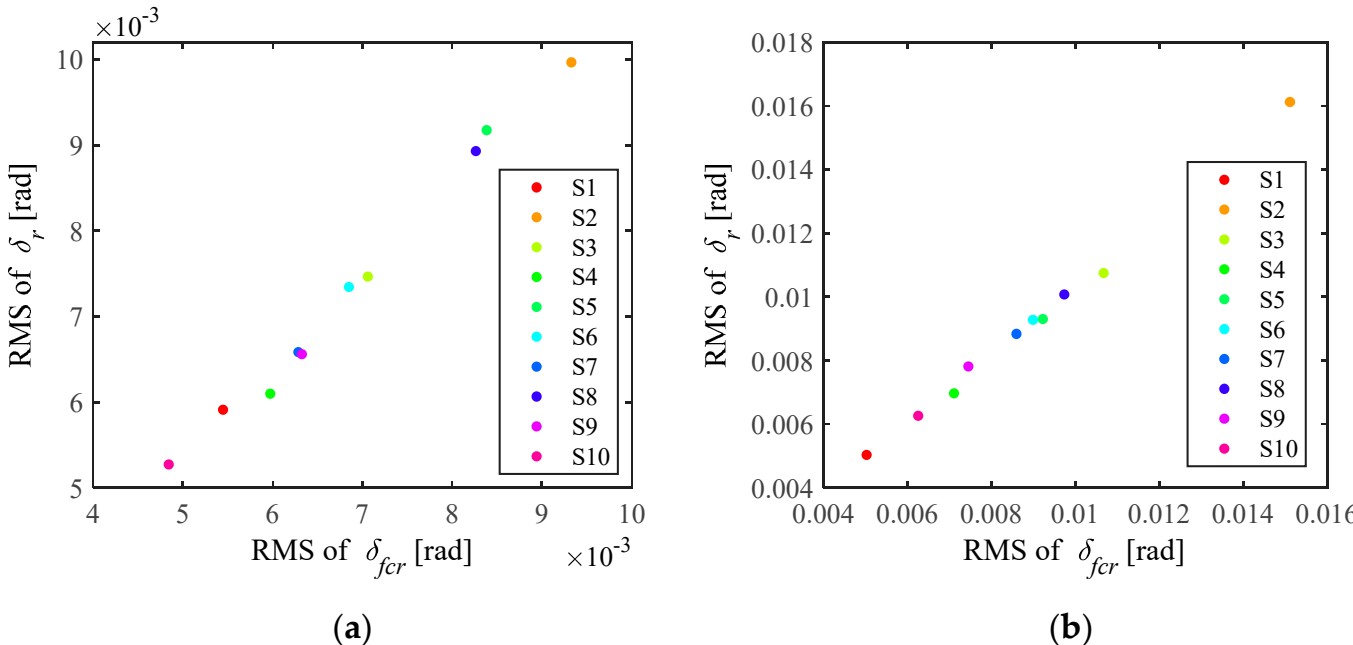

**Figure 24.** Correlation diagrams between the RMS of the front steering angle deviation and the RMS of the rear steering angle at (**a**) $V$ = 60 km/h and (**b**) $V$ = 80 km/h.

## 5. Conclusions

In this study, we defined the risk potential field for emergency avoidance driving based on road environment information and proposed a unified control law of a rear-wheel steering system that contributed to human-centered autonomous driving for active safety technology. The major conclusions obtained from the analysis of the simulations and experiments can be summarized as follows:

(1) The vehicle could generate a yaw rate equivalent to that of the reference driver and derive a safe and smooth trajectory in a double lane change test by applying the risk potential field 4WS.

(2) The experiments were conducted three times under each condition, and the trajectory variation became smaller when the risk potential field 4WS was applied.

(3) The corrective steering and steering burden of the driver were reduced by applying the risk potential field 4WS in comparison with the 2WS and conventional 4WS. Consequently, the handling quality and emergency avoidance performance were enhanced.

(4) The amount of steering support for the rear wheels in the proposed method was adjusted while considering the driving characteristics. Hence, the system could guide the driver to safety while maintaining the driver's vehicle control authority.

(5) In this study, the simulation was conducted under the assumption that the risk potential field was generated accurately from the road environment information. However, for practical use, it was necessary to conduct experimental verification while actually obtaining information from the sensors and maps in real time. In addition, there were obstacles such as pedestrians and other vehicles in the actual driving environment. Therefore, if the effectiveness of the proposed control could be confirmed after defining the obstacle potential, its feasibility would be further enhanced.

In the future, the application of the proposed risk-sensitive 4WS system in the other driving scenarios to support driving safety will be studied, and the driver–vehicle interaction and shared control characteristics will be discussed. As for recent deployment of 4WS-equipped production vehicles [14], experimental tests in real vehicles will also be investigated in the near future.

This study has shown one example of adding an active chassis control system in the vehicle for enhancing automated driving functions (e.g., lane keeping and steering assistance for collision avoidance). Aside from the four-wheel steering system, the four-wheel drive system (4WD) is also one of the promising chassis control technologies which can be used in advanced vehicle automation with high stability and vehicle dynamics capability [23–36]. Especially in the era of vehicle electrification, further developments of the active chassis control applied to vehicle automation will be intensively studied in the near future.

**Author Contributions:** Conceptualization, T.K.; formal analysis, T.K. and P.R.; funding acquisition, P.R.; methodology, T.K. and P.R.; software, T.K.; supervision, T.K.; validation, T.K.; visualization, T.K.; writing—original draft, T.K. and P.R.; writing—review and editing, P.R. All authors have read and agreed to the published version of the manuscript.

**Funding:** This research received no external funding.

**Institutional Review Board Statement:** Not applicable.

**Informed Consent Statement:** Not applicable.

**Data Availability Statement:** Not applicable.

**Conflicts of Interest:** The authors declare no conflict of interest.

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
