# Peer review of "Risk-Sensitive Rear-Wheel Steering Control Method Based on the Risk Potential Field"

_applsci, doi:10.3390/app11167296_

Round 1
Reviewer 1 Report
1. The introduction is not sufficient. The current advances in active steering and its control method should be investigated. For example:
[r1] Hierarchical Synchronization Control Strategy of Active Rear Axle Independent Steering System, applied science.
[r2] Robust active steering control for vehicle rollover prevention. International Journal of Modelling, Identification and Control.
[r3] Adaptive second order recursive terminal sliding mode control for a four-wheel independent steer-by-wire system. IEEE Access.
2. The contribution of this work should be emphasized in Introduction.
3. In sections 2.4-2.5,how to guarantee that the proposed closed-loop system is stable? Some essential proof or explanation is required.
4. what does the "risk" exactly mean in this paper? How to guarantee that the proposed "Risk Potential Field" is suitable and reasonable? Do you think the definition in (1)-(3) is more like a control precision instead of a risk index? I mean, the "Risk Potential Field" should be somewhat like rollover, crash, etc. In addition, the terms gamma, sigma are not defined.
5. In Section, 2.2, what is the constraint and the solver in the optimization problem?
6. The risk potential U_risk should also be ploted in the Simulation/ Experiment sections such that the readers can evaluate the advantages in a more illustrative way.
7. In fact, I think this work is not presented in a rigorous and clear way and I recommend the author to improve them in the next round of review.
Reviewer 2 Report
The article is interesting and fits in the global trend of automotive development. The manuscript may be of interest to a narrow readership. The manuscript contains theoretical considerations supported by an analytical method and a simulation test.
However, a few elements are missing:
- no reference to the results of research by teams from North America and Europe. It is obligatory to extend the introduction and add articles describing the research effects of other research centers;
- no reference in the introduction to the authors' own research (previous publications) to show that the article is the continuity of their research;
- no justification as to why the authors took up this subject. What results of own research indicated that it is worth taking care of this topic?
- here comes the question: why no one dealt with rear-wheel drive. Maybe 4WD is better and the trends in automotive development indicate that in the future cars will only have 4WD because they are safer.
Conclusions
(2) - I think that after 3 simulations test it cannot be concluded that the proposed method is "highly repeatable”
(4) - please develop what research plans authors have - especially what scope of real research is planned.
Round 2
Reviewer 1 Report
The reply is not convincing. The authors should carefully consider the following problems:
- In question 3,the explanation is not sufficient. I think a mathematical or a more rigorous explanation is required, or, at least, give the complex plane of the closed-loop system.
- The added sentence "The reference yaw rate increment/decrement candidate ranges between -0.1 and 0.1 rad/s." is not enough. This constraint MUST be described in mathematical structure as that in a standard optimal problem. In addition, the solover, I mean Runge kutta, Euler or other solvers?
- Fig 5 is just the "Risk potential field " and not the outcome of the simulation and experiment examples of the paper.
- In equations, Y is recommended to replaced by a standard form to prevent a confusion with γ.
- I find the that recommended reference [r2] is not cited although I think that rollover is a classic risk in vehicle control which is relative to the background of this paper.
- Is the constraint a hard one or a soft one? I mean, although the constraint of the lateral displacement is constrained in the optimal problem, is there any possibility that the vehicle still break through the constraint, especially if the control parameters are not suitably selected? At this case, the vehicle is still dangerous. I think this phenomenon should be discussed in suitable place. In addition, the control parameters used in the simulation and experiment should be added.
Reviewer 2 Report
No comments
Author Response
We appreciate that you reviewed our revised manuscript. We revised manuscript again according to the comments from Reviewer 1. So we wish if you would read the our revised manuscript. Thank you.